# Benford's Law applies to word frequency rank in English, German, French, Spanish, and Italian

**Jennifer Golbeck** [ORCID] *

University of Maryland, College Park, MD, United States of America

* jgolbeck@umd.edu

## Abstract

Benford's Law states that, in many real-world data sets, the frequency of numbers' first digits is predicted by the formula $log(1 + (1/d))$. Numbers beginning with a 1 occur roughly 30% of the time, and are six times more common than numbers beginning with a 9. We show that Benford's Law applies to the the frequency rank of words in English, German, French, Spanish, and Italian. We calculated the frequency rank of words in the Google Ngram Viewer corpora. Then, using the first significant digit of the frequency rank, we found the FSD distribution adhered to the expected Benford's Law distribution. Over a series of additional corpora from sources ranging from news to books to social media and across the languages studied, we consistently found adherence to Benford's Law. Furthermore, at the user-level on social media, we found Benford's Law holds for the vast majority of users' collected posts and significant deviations from Benford's Law tends to be a mark of spam bots.

## Introduction

Benford's Law states that, in many real-world data sets, the frequency of numbers' first digits is not evenly distributed. The exact frequency $P$ predicted for a first significant digit (FSD) $d$ is given by this formula:

$$P(d) = \log_{10}(1 + (1/d))$$

Benford's Law is most widely used in forensic accounting, where a violation may indicate fraud [1]. Research has also shown that it applies to Covid reporting [2, 3], genome data [4], scientific regression coefficients [5], the stock market in some cases [6] but not others [7], and can be used to detect AI-generated images [8]. Generally, if Benford's Law typically applies to a system, deviations from it can be indicators of "unnatural" activity.

In statistical linguistics, both word frequency (the number of times a word appears in a corpus) and word *frequency rank* (which word is most popular, 2nd most popular, and so on) are common measures. For example, these are the ten most popular words with their frequency rank and overall occurrence in an English corpus:

1. the (53,097,401,461)

**Data Availability Statement:** We used 23 public corpra in this study Google - English http://storage. googleapis.com/books/ngrams/books/datasetsv2. html Version 20120701 Google - German http:// storage.googleapis.com/books/ngrams/books/

datasetsv2.html Version 20120701 Google - Spanish http://storage.googleapis.com/books/ngrams/books/datasetsv2.html Version 20120701 Big English https://www.kaggle.com/datasets/rtatman/english-word-frequency?resource=download Google - French http://storage.googleapis.com/books/ngrams/books/datasetsv2.html Version 20120701 Google -Italian http://storage.googleapis.com/books/ngrams/books/datasetsv2.html Version 20120701 French Reddit https://www.kaggle.com/datasets/breandan/french-reddit-discussion Cresci http://mib.projects.iit.cnr.it Twitter abortion https://www.kaggle.com/datasets/mcantoni81/italian-tweets-discussion-on-the-roevswade LELU https://www.kaggle.com/datasets/breandan/french-reddit-discussion Spanish Wikipedia https://www.kaggle.com/datasets/rtatman/120-million-word-spanish-corpus COW https://www.webcorpora.org/opendata/frequencies/ Aranea corpra http://unesco.uniba.sk/aranea_about/ DeReKo https://www.corpusfinder.ugent.be/corpus/69 BNC http://www.kilgarriff.co.uk/bnc-readme.html parler https://zenodo.org/record/4442460.

**Funding:** The author(s) received no specific funding for this work.

**Competing interests:** The authors have declared that no competing interests exist.

2. of (30,966,074,232)

3. and (22,632,024,504)

4. to (19,347,398,077)

5. in (16,891,065,263)

6. a (15,310,087,895)

7. is (8,384,246,685)

8. that (8,000,768,228)

9. for (6,545,282,031)

10. it (5,740,085,369)

For this research, we measured frequency and frequency rank using the Google 1-gram datasets for English, German, French, Spanish, and Italian. Described further below, these corpora contain the count of how often every work appears in a large collection of around 8 million books published between 1500 and 2008, (roughly 6% of all books ever published).

Then, we looked at the distribution of frequency rank FSDs within a variety of additional corpora. We found across corpora and languages, these distributions closely followed Benford's expected distribution and were statistically indistinguishable. Furthermore, the FSD distribution of the frequency rank of words posted by individual users also closely follow Benford's Law. In two corpora, we show that when a user's posts deviate from Benford's Law, it is associated with spam bot behavior. This echoes previous work that shows violations of Benford's Law in social network connection behavior is also an indicator of bot activity [9].

We discuss how these results lead to the possibility of using Benford's Law to detect bots and other unusual or unnatural behavior.

## Background and related work

### Benford's Law

Benford's Law was first detected by astronomer Simon Newcomb and later validated by physicist Frank Benford. Benford studied naturally occurring numbers from many sources: the surface area of rivers, atomic weights, and numbers appearing in Reader's Digest, and found all followed the same FSD distribution [10].

The formula for the law, $P(d) = \log_{10}(1 + (1/d))$, provides a theoretical distribution of expected first digits, shown in Table 1.

Benford's Law is frequently applied in financial and accounting applications, specifically to detect fraud [11]. However, Benford's law is not always present even when researchers may expect it to be based on the form of the data. For example, [7] examined S&P500 daily closing values and found non-conformity with Benford. Thus, it is important to apply and test with Benford using a careful eye.

Benford's Law often applies to systems that follow a power law distribution [12]. Power laws are commonly found in social network structures [13] social media [14], and online behavior [15]. In our previous studies, we have shown that Benford's Law applies to friend and

**Table 1. Frequency of first significant digits (FSD) expected by Benford's Law.**

| FSD | 1 | 2 | 3 | 4 | 5 | 6 | 7 | 8 | 9 |
|---|---|---|---|---|---|---|---|---|---|
| Frequency | 0.301 | 0.176 | 0.125 | 0.097 | 0.07918 | 0.067 | 0.05799 | 0.051 | 0.046 |

follower counts in social media across platforms, both at the platform level and the individual level [16]. We also showed that violations of Benford's Law in network structure can be used to identify malicious bots [9].

## Rank-frequency distribution, Zipf's Law, and language

The frequency of a word in a corpus is the number of times it occurs. The rank-frequency, or frequency rank, is the rank of a word when the frequencies of all words are sorted from most to least frequent. The most frequent word in a corpus would have a rank of 1. Frequency rank distributions, not just in languages but of most data, tend to follow a power law distribution [17]. Power laws appear frequently in nature and occur when one variable varies as a power of the other.

Zipf's law is a particular type of power law distribution that states that frequency is inversely proportional to rank. It was originally formulated with respect to linguistic analysis, addressing the frequency of words in a corpus [18]. Researchers have found that word frequency adheres to Zipf's law [19].

Zipf's Law and Benford's Law have some relationship. Studies have shown that in some contexts, Benford's Law and Zipf's law both apply, but not all Zipfian distributions follow Benford's Law [12].

Despite Zipf's Law originating from the study of language, and a statistical connection between Benford's Law and Zipf's law, there is almost no research into Benford's Law applying to language. One study found that Chinese character and word frequency could be described by Benford's Law on small-scale texts [20]. However, there are no studies on large corpora, that look at other languages, or that address frequency rank and Benford's Law.

## Methods and data

Our foundational question in this research is if Benford's Law applies to word frequency rank across languages. If Benford's Law consistently holds across diverse datasets in a given language, it is evidence that we can expect Benford to hold for most (but not necessarily all!) datasets in that language. That, in turn, would indicate that we may be able to use Benford testing to look for "abnormal" behavior which, in turn, could be an initial indicator of fraudulent or manipulated behavior.

The benefit of using frequency rank is that it allows comparison across corpora. The frequency count of the word "that", for example, will vary in every corpus, but it's rank should be approximately the same. In this analysis, using frequency rank also allows us to establish a standard frequency rank for the language as a whole, using very large datasets, which we can then use to evaluate the use in each corpus based on the same values. For example, "that" will be ranked 8 in English and considered the 8th most frequent English word in every corpus, rather than re-measuring it within each corpus.

## Establishing word frequency rank

The frequency of a given word in a language can never really be measured since, for example, the vast majority of words spoken are never recorded. Even in recorded language, frequency depends on many factors, including time period, context, region, and format (e.g. written or spoken). It must be measured within a corpus, and corpora vary with respect to all of these factors. For this research, we opted to use the largest corpora available to establish a baseline frequency and frequency rank for words.

We used the Google Books Ngram Viewer dataset [21]. These datasets are built from a large collection of around 8 million books published between 1500 and 2008, which account for

roughly 6% of all books ever published. The English dataset is the largest, with approximately 1 trillion words. The German, Spanish, and French datasets each have over 100 billion words, and the Italian dataset has over 60 billion. These are orders of magnitude larger than any other available corpora.

Google makes frequency counts available for each word. To calculate frequency and frequency rank, we first removed tokens that were not words (numbers, punctuation) and then sorted by frequency. We assigned each word a frequency rank beginning at 1 for the most popular word (e.g. "the" in the English corpus).

Analyzing one dataset is not enough to make claims about general rules for language use [22]. In this case, if English in one dataset follows a Benford's Law distribution, that does not necessarily mean we can draw conclusions about language or even English generally. To mitigate these concerns, we followed the guidelines of [21], considering multiple languages and validating our results on different datasets in the target languages.

We applied the same ranking process to the Google datasets for German, French, Spanish, and Italian. We used these frequency ranks for our Benford's Law analysis of each language. We also used these frequency ranks as a basis for analyzing other corpora in each language to see if the Benford's distribution was observable in smaller datasets drawn from a variety of sources. These corpora are discussed in the next section.

## Additional corpora

To check if Benford's Law applies to word usage with respect to frequency rank, we tested on a number of additional large corpora across all five languages. These are drawn from news, social media, books, and the web. The number of total words and unique words in each dataset are given in Tables 2–6. The purpose of this analysis was to validate results obtained from the Google Ngrams analysis on very diverse corpora. If Benford's Law applies robustly to language use, it should be observable in many contexts, corpora of many sizes, and from a variety of sources. Thus, the purpose in choosing these was not to select the biggest or most balanced or those from a specific source. Rather, they were chosen to test how well Benford works in a range of contexts and languages.

Many of these corpora contain non-standard words like usernames, hashtags, misspellings, and URLs. They also have "words" which are numbers, codes, punctuation, emoji, etc. We

**Table 2. Distribution of FSD of word's frequency rank based on use in Google Ngram Viewer corpora.** Total word counts are measured in billions (B).

| FSD | Benford | English | German | Spanish | Italian | French |
|---|---|---|---|---|---|---|
| 1 | 0.30 | 0.32 | 0.30 | 0.33 | 0.32 | 0.32 |
| 2 | 0.18 | 0.18 | 0.19 | 0.17 | 0.18 | 0.18 |
| 3 | 0.12 | 0.13 | 0.13 | 0.11 | 0.12 | 0.12 |
| 4 | 0.10 | 0.10 | 0.10 | 0.09 | 0.09 | 0.09 |
| 5 | 0.08 | 0.08 | 0.07 | 0.08 | 0.08 | 0.08 |
| 6 | 0.07 | 0.07 | 0.06 | 0.07 | 0.06 | 0.07 |
| 7 | 0.06 | 0.05 | 0.05 | 0.06 | 0.06 | 0.06 |
| 8 | 0.05 | 0.04 | 0.05 | 0.05 | 0.05 | 0.05 |
| 9 | 0.05 | 0.04 | 0.04 | 0.04 | 0.04 | 0.04 |
| | Total Words | 755 B | 103 B | 107 B | 64 B | 167 B |
| | Unique words | 4,999,751 | 4,540,689 | 2,250,889 | 1,825,177 | 2,741,602 |
| | Pearson $\rho$ | 1.000 | 0.998 | 0.996 | 0.999 | 0.999 |
| | K-S value | 0.222 | 0.111 | 0.111 | 0.111 | 0.111 |
| | K-S p-value | 0.9895 | 1 | 1 | 1 | 1 |

**Table 3. Distribution of FSD of word's frequency rank based on use in additional English corpora.**

| FSD | Benford | Parler | Cresci | BNC | Leipzig |
|---|---|---|---|---|---|
| 1 | 0.30 | 0.30 | 0.29 | 0.31 | 0.31 |
| 2 | 0.18 | 0.15 | 0.17 | 0.17 | 0.16 |
| 3 | 0.12 | 0.14 | 0.13 | 0.13 | 0.13 |
| 4 | 0.10 | 0.11 | 0.11 | 0.10 | 0.10 |
| 5 | 0.08 | 0.07 | 0.07 | 0.08 | 0.08 |
| 6 | 0.07 | 0.07 | 0.07 | 0.07 | 0.07 |
| 7 | 0.06 | 0.06 | 0.07 | 0.05 | 0.05 |
| 8 | 0.05 | 0.05 | 0.04 | 0.05 | 0.05 |
| 9 | 0.05 | 0.05 | 0.04 | 0.04 | 0.04 |
| | Total Words | 1,760,010,838 | 73,520,025 | 95,168,106 | 17,310,077 |
| | Unique words | 1,848,342 | 620,828 | 927,596 | 417,396 |
| | Pearson $\rho$ | 0.989 | 0.993 | 0.998 | 0.997 |
| | K-S Value | 0.111 | 0.222 | 0.222 | 0.111 |
| | K-S p-value | 1 | 0.9895 | 0.9895 | 1 |

applied a high level filter to remove numbers and punctuation. Other non-standard words that did not appear in the Google datasets were ignored. While this can constitute a significant percentage of the *unique* words in a corpus, it usually reflects a small percentage of the *total* words. For example, in the Parler dataset, there were 1,848,342 unique words but only 690,590 were in the Google corpus. While it may appear that the majority of words were excluded from analysis, these excluded words made up only 0.79% of all the words used on Parler. Most were usernames and hashtags (e.g. jreidusmct, funnyquotes, ycgoiszxsvipw, oyrgdjxfgxsl, xfajndhbbzz, saddayforamerica). Thus, because the removed words are such a small percentage of all words used, their exclusion should have a negligible impact on the Benford's analysis. (See also [7] for an analogous discussion of the presence of zeroes in financial data).

- Cresci (English—Social Media)
  The Fake Project dataset is a dataset of Twitter bots and genuine accounts [23]. Across four categories of accounts, the dataset contains 6.6 million tweets.

**Table 4. Distribution of FSD of word's frequency rank based on use in additional German corpora.**

| FSD | Benford | DeRoKo | Aranea | COW | Leipzig |
|---|---|---|---|---|---|
| 1 | 0.30 | 0.31 | 0.30 | 0.30 | 0.32 |
| 2 | 0.18 | 0.19 | 0.19 | 0.19 | 0.18 |
| 3 | 0.12 | 0.13 | 0.14 | 0.14 | 0.13 |
| 4 | 0.10 | 0.10 | 0.10 | 0.10 | 0.09 |
| 5 | 0.08 | 0.08 | 0.08 | 0.08 | 0.08 |
| 6 | 0.07 | 0.06 | 0.06 | 0.06 | 0.06 |
| 7 | 0.06 | 0.05 | 0.05 | 0.05 | 0.05 |
| 8 | 0.05 | 0.04 | 0.04 | 0.05 | 0.04 |
| 9 | 0.05 | 0.04 | 0.04 | 0.04 | 0.05 |
| | Total Words | 7,301,006,311 | 100,379,842 | 11,331,629,659 | 15,089,080 |
| | Unique words | 98,072 | 483,408 | 71,092,150 | 838,937 |
| | Pearson $\rho$ | 0.999 | 0.997 | 0.997 | 0.999 |
| | K-S Value | 0.222 | 0.222 | 0.222 | 0.222 |
| | K-S p-value | 0.9895 | 0.9895 | 0.9895 | 0.9895 |

**Table 5. Distribution of FSD of word's frequency rank based on use in additional Spanish corpora.**

| FSD | Benford | COW | Aranea | LELU | Leipzig |
|---|---|---|---|---|---|
| 1 | 0.30 | 0.34 | 0.33 | 0.36 | 0.33 |
| 2 | 0.18 | 0.16 | 0.17 | 0.16 | 0.17 |
| 3 | 0.12 | 0.11 | 0.11 | 0.11 | 0.11 |
| 4 | 0.10 | 0.09 | 0.09 | 0.09 | 0.09 |
| 5 | 0.08 | 0.09 | 0.08 | 0.08 | 0.08 |
| 6 | 0.07 | 0.07 | 0.07 | 0.05 | 0.08 |
| 7 | 0.06 | 0.06 | 0.06 | 0.05 | 0.06 |
| 8 | 0.05 | 0.04 | 0.04 | 0.04 | 0.05 |
| 9 | 0.05 | 0.03 | 0.04 | 0.04 | 0.04 |
|  | Total Words | 3,163,716,419 | 97,268,117 | 75,709,619 | 19,817,544 |
|  | Unique words | 8,540,745 | 272,471 | 3,800,336 | 413,687 |
|  | Pearson $\rho$ | 0.989 | 0.995 | 0.991 | 0.994 |
|  | K-S Value | 0.222 | 0.222 | 0.222 | 0.111 |
|  | K-S p-value | 0.9895 | 0.9895 | 0.9895 | 1 |

- Parler (English—Social Media)

  Parler was a Twitter-like microblogging platform that became popular among Trump sup-porters and conservatives, particularly in 2020. It tagged itself as "Parler: Speak freely and express yourself openly without fear of being deplatformed for your views" and was seen as a haven after far-right figures were suspended from Twitter for violating their Terms of Service. Parler was a major planning platform for the January 6 insurrection in the United States and it was shut down within a few days of the attacks on the Capitol when their service providers broke ties.

  We analyzed a dataset of Parler posts that spans August 2018 to January 2021 and was collected and described in [24]. It includes 183 million posts from 4 million users. Most posts are in English, and we only considered the English language posts in our analysis.

- British National Corpus (English—wide coverage)

  The British National Corpus (BNC) [25] is a large corpus of samples of English, both written

**Table 6. Distribution of FSD of word's frequency rank based on use in additional French corpora.**

| FSD | Benford | Aranea | Reddit | Leipzig |
|---|---|---|---|---|
| 1 | 0.30 | 0.32 | 0.34 | 0.32 |
| 2 | 0.18 | 0.18 | 0.19 | 0.19 |
| 3 | 0.12 | 0.14 | 0.11 | 0.13 |
| 4 | 0.10 | 0.09 | 0.11 | 0.09 |
| 5 | 0.08 | 0.07 | 0.06 | 0.07 |
| 6 | 0.07 | 0.06 | 0.06 | 0.06 |
| 7 | 0.06 | 0.06 | 0.05 | 0.06 |
| 8 | 0.05 | 0.05 | 0.05 | 0.04 |
| 9 | 0.05 | 0.03 | 0.03 | 0.03 |
|  | Total Words | 104,152,457 | 66,518,066 | 19,419,591 |
|  | Unique words | 261,637 | 772,033 | 360,383 |
|  | Pearson $\rho$ | 0.997 | 0.995 | 0.998 |
|  | K-S Value | 0.111 | 0.222 | 0.222 |
|  | K-S p-value | 1 | 0.9895 | 0.9895 |

**Table 7. Distribution of FSD of word's frequency rank based on use in additional Italian corpora.**

| FSD | Benford | Aranea | Twitter—Abortion | Leipzig |
|---|---|---|---|---|
| 1 | 0.30 | 0.33 | 0.35 | 0.32 |
| 2 | 0.18 | 0.17 | 0.16 | 0.17 |
| 3 | 0.12 | 0.10 | 0.12 | 0.11 |
| 4 | 0.10 | 0.06 | 0.09 | 0.11 |
| 5 | 0.08 | 0.09 | 0.07 | 0.08 |
| 6 | 0.07 | 0.09 | 0.07 | 0.06 |
| 7 | 0.06 | 0.06 | 0.06 | 0.06 |
| 8 | 0.05 | 0.08 | 0.04 | 0.05 |
| 9 | 0.05 | 0.04 | 0.05 | 0.05 |
| | Total Words | 94,921,972 | 1,175,775 | 18,716,950 |
| | Unique words | 314,426 | 55,373 | 430,832 |
| | Pearson $\rho$ | 0.967 | 0.989 | 0.996 |
| | K-S Value | 0.111 | 0.111 | 0.222 |
| | K-S p-value | 1 | 1 | 0.9895 |

and spoken. It is a balanced corpus which is supposed be representative of how English is used. The written samples, which make up around 90% of the corpus, are collected from newspapers, magazines, journals, fiction and non-fiction books, letters, school essays, and other text. The spoken portion also seeks to use a wide ranging sample, including lectures, radio shows, call-in shows and classroom discussions.
We used the unlemmatized frequency list from the BNC provided at [26].

- The German Reference Corpus DeRoKo (German—books and newspapers)
  The German Reference Corpus or Deutsches Referenzkorpus (DeRoKo) [27] is a large reference corpus of German texts. It is built from fiction, scientific manuscripts, and newspapers. We conducted our analysis using the DeReKo-2014-II-MainArchive-STT.100000.freq list, which contains approximately 7 billion words.

- Aranea (German, Spanish, French, Italian—web pages)
  The Aranea corpora comprise web pages crawled beginning in 2013 [28–30]. We used the German, Spanish, French, and Italian corpora in our analysis.

- Leipzig News Corpus (German, Spanish, French, Italian, English) The Leipzig corpora are collected from newspaper text and released yearly for each language [31, 32]. We used the 1 million sentence corpora from 2008 for English, Spanish and German, and the integrated 2005-2008 corpus for French and Italian.

- Corpora from the Web—COW (German, Spanish—web pages)
  The Corpora from the Web project provides web corpora for multiple European languages, including German and Spanish [33, 34].

- Twitter Abortion Discussion (Italian—Social Media)
  This focused corpus of around 47,000 tweets was collected in June 2022 after the overturn of Roe v Wade. The tweets are in Italian and were selected if they contained "abortion", "aborto" or "abortire". We intentionally include this smaller and more focused dataset as a test of whether Benford's Law applies to word use in specific contexts.

- LÉLU French dialog corpus (French—Social Media)
  The LÉLU French dialog corpus [35] is a collection of 556,621 comments from French-language subreddits: /r/france, /r/FrancaisCanadien, /r/truefrance, /r/paslegorafi, and /r/rance.

- 120M Word Corpus (Spanish—Wikipedia)
  The 120M word Spanish Wikicorpus [36] is collected from Spanish-language Wikipedia pages.

## Results

### Benford's Law applies across corpora

To investigate the presence of Benford's Law in language, we considered words' frequency rank, as described above.

We then represented each word by the FSD of its frequency rank. In this case, both "the" (rank 1) and "it" (rank 10) would be represented as a 1, "of" would be 2, etc. We have frequency counts for each word, so we summed the frequencies for all words with each FSD to get the total occurrence of each FSD in the corpus. Dividing this by the total number of words in the corpus produces the frequency of each FSD.

For the Google corpora, we computed these values for each language. The frequency of each FSD is shown in Table 2. We first computed the Pearson correlation between the observed frequency and the frequency expected if Benford's Law applies. Pearson correlations are a common way to measure how closely a distribution adheres to Benford's Law [37–39].

As shown in Table 2, all correlations are almost perfect, with the lowest $\rho = 0.996$. All correlations were significant for $p < 0.001$ with a Bonferroni correction of 23 to account for all corpora correlations performed in this study.

We also ran Kolmogorov–Smirnov (K-S) tests to check the fit of the data with the Benford's Law distribution. The K-S $p$-values, which indicate the probability that the social network's FSD distribution is the same as Benford's, are also shown in Table 2. These values are all $> 0.98$, indicating a near perfect match with the expected distribution.

We performed the same analysis on additional corpora using the frequency ranks from the Google dataset. As with the large Google corpora, the Pearson correlation was strong on every additional corpus. Every value was $> 0.96$ and all were significant for $p < 0.001$ with the Bonferroni correction described above. Similarly, the K-S tests showed strong adherence to Benford's Law, with p-values $> 0.98$. Results for each language are shown in Tables 3 to 7.

### User-level analysis on social media

In previous studies, we showed that Benford's Law applied to the network structure of social media platforms as a whole and also to individual users' egocentric networks [16]. When individual users significantly deviated from the expected Benford distribution, it was often an indicator that the accounts were bots. This inspired similar questions for this analysis. Does Benford's Law hold for the frequency rank of individual users' word choice and, if so, does deviation from Benford indicate bot-like behavior?

Among the social media corpora considered in this analysis, Parler and Cresci had many tweets per individual user. The French Reddit and Italian Twitter corpora often had only one post per user which was not enough text to conduct this type of analysis.

The Cresci data has an additional benefit in that it is a labeled dataset for analyzing bots. Accounts are labeled as "genuine" or one of several types of bots. We considered genuine accounts, social spam bots, and traditional spam bots.

For each user in these groups, we built a set of their tweets. We excluded any users who had posted fewer than 1,000 words to ensure our statistics would be meaningful. This left us with 1,048 genuine accounts, 2,214 social spam accounts, and 160 traditional spam accounts.

For each user, we measured the distribution of FSDs of the frequency rank of each word they used. We then compared each individual user's FSD distribution against the expected Benford distribution. As is common practice, we used the Pearson correlation to measure how well the distributions fit.

The mean correlation coefficient for genuine accounts (0.977) was significantly higher than that of social spam bots (0.947) or traditional spam bots (0.908). A one-way ANOVA revealed that there was a statistically significant difference in the Pearson correlation coefficients between at least two groups ($F(2, 3419) = [281.1596]$, $p = 0$). Tukey's HSD Test for multiple comparisons found that the mean correlation coefficient was significantly different between genuine users and social spam bots ($p = < 0.001$, $95\% C.I. = [0.061, 0.078]$) and between genuine users and traditional spam bots ($p = < 0.001$, $95\% C.I. = [0.026, 0.033]$).

In the Parler dataset, accounts are not labeled as bots or not. As a preliminary investigation, we sorted accounts by their Pearson $\rho$ and qualitatively analyzed 100 accounts—the 50 with the strongest correlations and the 50 accounts with the weakest correlations. Based on visual inspection of the content, there did not appear to be any spam bots among the top Benford-correlated Parler users. Among the 50 weakest correlates, inspection of their posts indicated 45 were obvious spam bots and 3 more were bots posting other automated content. Among the obvious spam bots were accounts that posted the same message over and over. These are representative examples of the repeated posts from five different accounts:

- Welcome! Follow me for conservative Memes and News.

- Welcome to Parler, Patriot! Follow me and I will definitely give you a follow back!

- Follow me for funny conservative comedy on parler Don't forget to echo!

- Follow me for more Trump tweets! GOD bless all you Patriots!

- The White Rabbit welcomes you to Parler. Hope to see you around the 🐰 hole! Follow the White Rabbit. . .

## Discussion and conclusions

We have shown that Benford's Law applies to the frequency rank distribution of words in English, French, German, Spanish, and Italian. Using large datasets from a range of sources, we consistently found a strong correlation and statistical adherence to the expected Benford's Law distribution. We also showed that Benford's Law applies to the smaller samples of individuals' social media posts in the vast majority of cases. However, spam bots tend to have a lower correlation with Benford's distribution and strong deviations from the expected distribution are indicators of bot activity.

While many bots *do* closely adhere to the FSD distribution predicted by Benford's Law, analysis for the Cresci bot dataset shows that there is some signal present indicating that a deviation from the expected distribution is tied to bot status. Furthermore, among the biggest deviators in the Parler dataset, almost all were spam bots. While correlation with Benford alone is not strong enough to indicate an account is a bot or not, these results suggest this could be one factor to include in bot detection algorithms to potentially improve their performance.

These results suggest a number of avenues for future work. First, given there is some relationship between data that follows Zipf's Law and Benford's Law [12], Benford tests can be expanded further. We have focused on Western European languages and a natural next step would be to verify this in a wider range of languages. Existing work suggests this likely would work, given the word frequency results in Chinese [20]. Zipf's Law has also been shown to hold in the vocalizations of other species, including dolphins [40] and penguins [41]; does Benford's Law apply as well? Additionally, Zipf himself posited psychological theories for why his law works. He that the relationship exists to minimize effort both for the speaker and hearer. The speaker wants to use the fewest number of words, while the hearer wants the words with the clearest meaning used [42]. There is an intriguing line of work to connect psychological research on Zipf's Law and language with Benford's Law to see if there are insights to be uncovered that can psychologically explain Benfordian behavior.

## Author Contributions

**Conceptualization:** Jennifer Golbeck.

**Data curation:** Jennifer Golbeck.

**Formal analysis:** Jennifer Golbeck.

**Investigation:** Jennifer Golbeck.

**Methodology:** Jennifer Golbeck.

**Project administration:** Jennifer Golbeck.

**Resources:** Jennifer Golbeck.

**Software:** Jennifer Golbeck.

**Validation:** Jennifer Golbeck.

**Writing – original draft:** Jennifer Golbeck.

**Writing – review & editing:** Jennifer Golbeck.

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
