## [Decision Letter · Decision Letter 0]

1 Jun 2023

PONE-D-23-08610Benford's Law Applies to Word Frequency Rank in English, German, French, Spanish, and ItalianPLOS ONE

Dear Dr. Golbeck,

Thank you for submitting your manuscript to PLOS ONE. After careful consideration, we feel that it has merit but does not fully meet PLOS ONE’s publication criteria as it currently stands. Therefore, we invite you to submit a revised version of the manuscript that addresses the points raised during the review process.

We look forward to receiving your revised manuscript.

Kind regards,

Luca De Benedictis, PhD

Academic Editor

PLOS ONE

Journal Requirements:

Reviewers' comments:

Reviewer's Responses to Questions

**Comments to the Author**

1. Is the manuscript technically sound, and do the data support the conclusions?

Reviewer #1: Yes

2. Has the statistical analysis been performed appropriately and rigorously? 

Reviewer #1: Yes

3. Have the authors made all data underlying the findings in their manuscript fully available?

Reviewer #1: Yes

4. Is the manuscript presented in an intelligible fashion and written in standard English?

Reviewer #1: Yes

5. Review Comments to the Author

Reviewer #1: PONE-D-23-08610

Reviewer comments

The present paper demonstrates that Benford's Law applies to ranked word frequencies in English, German, French, Spanish, and Italian. The author(s) computed the word frequency ranks using the Google Ngram Viewer corpus to prove this. By examining the first significant digit of the frequency ranks, the author(s) shows that the distribution of frequencies adheres to the expected pattern outlined by Benford's Law. Furthermore, across various sources such as news, books, and social media the author(s) investigated, we consistently found conformity to Benford's Law. Moreover, the authors analysed users' posts on social media and discovered that the vast majority of collected posts followed Benford's Law, while significant deviations from this pattern often indicated the presence of spam bots.

Introduction

In referring to Benford’s law studies in general and specifically on the stock market, more recent works may help contextualise. For example, Ausloos, M., Ficcadenti, V., Dhesi, G., & Shakeel, M. (2021). Benford’s laws tests on S&P500 daily closing values and the corresponding daily log-returns both point to huge non-conformity. Physica A: Statistical Mechanics and its Applications, 574, 125969.

The following statement should be simplified in favour of readers not familiar with Google 1-gram corpora. Furthermore, it is unclear what is intended by word’s frequency rank; namely, have you applied Benford’s law on the ranks or the frequencies? It may become clearer later in the text, but I feel it is important to say it here. Finally, what is “gold standard” in this context?

“We conducted an analysis of language use in a variety of corpra. To determine a word's frequency rank (e.g. what is the 314th most popular word in English), we measured frequency using the Google 1-gram corpra (reviewer’s note: CORRECT THE TYPO HERE) for English, German, French, 13 Spanish, and Italian. These frequency ranks were used as the gold standard rank in each language.”

A similar comment on the clarity of what “rank” and what “frequency” are under investigation should be applied to the paragraph related to social media posts.

Background and Related Work

This section could be improved by adding a practical example of the type of analysis that can be done with Benford’s law in linguistics. For example, mentioning a word, its rank in a corpus, and the way to consider its frequency. This comment is connected with the one made above about clarity.

Establishing Word Frequency Rank

One reads, “Google makes frequency counts available for each word. To calculate frequency and frequency rank, we first removed tokens that were not words (numbers, punctuation) and then sorted by frequency. We assigned each word a frequency rank beginning at 1 for the most popular word (e.g. “the" in the English corpus).” taken that a corpus is a book and that a corpora is a collection of books in a given language, does one have a rank for each word in every corpus? So, for example, one has N English books, each having a number of words and associated frequencies. I understand from the Results sections that words are considered altogether. Namely, one gets “approximately 1 trillion words” with their frequencies and ranks for English, correct? If so, I suggest to clarify here.

Additional Corpra

When the author(s) starts this section, the analysis and its results are not presented yet, but the basement of the sub-section is a need to validate the results. This is understandable, but a more thoughtful discussion of the potential results and their generalisability is necessary before introducing a “validation” argument.

In the following statement, “If Benford's Law applies robustly to language use, it should be observable in many contexts, corpra of many sizes, and both balanced and unbalanced corpra.” what does “balanced” mean?

Results

In this section, one finally understands, with an example, the concept of “frequency rank”. The usage of this approach is partially justified in the section “Rank-frequency distribution, Zipf's Law, and Language” but it is not enough because the obvious thought I had by reading is “why not use the frequencies directly?” I like the idea of using the ranking, so I am not discouraging, but some more on the reasons, the benefit of it, might be beneficial.

User-Level Analysis on Social Media

The methodology here must be clarified. Is the analysis based again on the “frequency rank”?

I do not get if the conclusion related to the fact that “bot” social network accounts are less Benford’s law compliant comes from a visual inspection or if it is possible to have some statistical test to validate that point.

Conclusions:

If the author(s) decides to keep any of the comments on board, please revise the conclusion accordingly. Furthermore, I suggest that, given the quality of the results in the field, more academically informed conclusive remarks can be given on the areas covered: the linguistic aspect and the social network component.

General comments:

When one removes non-standard text, I refer, e.g. to “Other non-standard words are unlikely to appear in the Google corpra and will be ignored” there might be a “manipulation” of the data that somewhat connect to what is done in the case of accountancy when some figures are systematically changed. Therefore, extra care should be used. Some of those words might be neologisms, words that are outside current dictionaries but in use in “hurban” conversation; this is especially relevant for the component of the analysis related to social networks. My question would be, does consider words in official dictionaries or including those not-in-dictionaries would change the results drastically? If it is not possible to re-do the analysis, I understand. I am not necessarily asking for that, but a few comments on the impact on Benford’s law validity would help. This is partially there already when the %s are reported, but a clear statement where one says that these numbers are not enough for breaking Benford’s law may reassure a reader. For example in “Ausloos, M., Ficcadenti, V., Dhesi, G., & Shakeel, M. (2021). Benford’s laws tests on S&P500 daily closing values and the corresponding daily log-returns both point to huge non-conformity. Physica A: Statistical Mechanics and its Applications, 574, 125969” a point regarding the inclusion or exclusion of the “zeros” from the returns analysis is made when evaluating the second digit law.

Check the usage of the words “corpora” and “corpus” I feel sometimes the usage of the words is interchanged, but the meaning is different.

Congratulations on using genuine and non-genuine accounts for the analysis related to the bots! I really like it!

6. PLOS authors have the option to publish the peer review history of their article (what does this mean?). If published, this will include your full peer review and any attached files.

Reviewer #1: No

---

## [Author Response · Author response to Decision Letter 0]

2 Jun 2023

(this response is also included as a submission document)

Response to Reviewers

Thank you so much for these very helpful comments. They have allowed us to make the paper much better and have been a tremendous help to the clarity. Below we have listed the reviewer comments in black and our responses in blue. We made all the changes requested by the reviewers.

Introduction

In referring to Benford’s law studies in general and specifically on the stock market, more recent works may help contextualise. For example, Ausloos, M., Ficcadenti, V., Dhesi, G., & Shakeel, M. (2021). Benford’s laws tests on S&P500 daily closing values and the corresponding daily log-returns both point to huge non-conformity. Physica A: Statistical Mechanics and its Applications, 574, 125969.

Thank you for this reference. We have added this to the introduction and also as a useful reference in the background to point out that Benford's law does not simply always apply in this type of data.

The following statement should be simplified in favour of readers not familiar with Google 1-gram corpora. Furthermore, it is unclear what is intended by word’s frequency rank; namely, have you applied Benford’s law on the ranks or the frequencies? It may become clearer later in the text, but I feel it is important to say it here. Finally, what is “gold standard” in this context?

“We conducted an analysis of language use in a variety of corpra. To determine a word's frequency rank (e.g. what is the 314th most popular word in English), we measured frequency using the Google 1-gram corpra (reviewer’s note: CORRECT THE TYPO HERE) for English, German, French, 13 Spanish, and Italian. These frequency ranks were used as the gold standard rank in each language.”

A similar comment on the clarity of what “rank” and what “frequency” are under investigation should be applied to the paragraph related to social media posts.

We have clarified this sentence, added background on the 1-gram corpora, and dropped "gold standard" from the paragraph for clarity.

We also moved the entire frequency rank discussion up to the introduction to address the many points about clarity raised by the reviewer.

Background and Related Work

This section could be improved by adding a practical example of the type of analysis that can be done with Benford’s law in linguistics. For example, mentioning a word, its rank in a corpus, and the way to consider its frequency. This comment is connected with the one made above about clarity.

This point should be addressed by the addition of the frequency rank discussion to the introduction.

Establishing Word Frequency Rank

One reads, “Google makes frequency counts available for each word. To calculate frequency and frequency rank, we first removed tokens that were not words (numbers, punctuation) and then sorted by frequency. We assigned each word a frequency rank beginning at 1 for the most popular word (e.g. “the" in the English corpus).” taken that a corpus is a book and that a corpora is a collection of books in a given language, does one have a rank for each word in every corpus? So, for example, one has N English books, each having a number of words and associated frequencies. I understand from the Results sections that words are considered altogether. Namely, one gets “approximately 1 trillion words” with their frequencies and ranks for English, correct? If so, I suggest to clarify here.

Google refers so their 1 Trillion Word dataset as a single corpus, rather than a corpora of all the books. After reviewing your comments (including one about corpora vs corpus below) think this is likely the source of the confusion. To clarify, we've updated the text to refer to the google data as a "dataset" (since it has more data in it than just words, anyway). 

Additional Corpora

When the author(s) starts this section, the analysis and its results are not presented yet, but the basement of the sub-section is a need to validate the results. This is understandable, but a more thoughtful discussion of the potential results and their generalisability is necessary before introducing a “validation” argument.

We have added this discussion to the beginning of the Methods and Data section, just above Establishing Word Frequency Rank.

In the following statement, “If Benford's Law applies robustly to language use, it should be observable in many contexts, corpra of many sizes, and both balanced and unbalanced corpra.” what does “balanced” mean?

"Balanced" corpora are intended to reflect overall usage, including speaking, fictional writing, non-fiction, etc. Honestly, it's only a concept we encountered in the process of doing this work and is not critical to the paper, so we've updated this sentence to instead refer to data "from a variety of sources" which captures the idea without the jargon.

Results

In this section, one finally understands, with an example, the concept of “frequency rank”. The usage of this approach is partially justified in the section “Rank-frequency distribution, Zipf's Law, and Language” but it is not enough because the obvious thought I had by reading is “why not use the frequencies directly?” I like the idea of using the ranking, so I am not discouraging, but some more on the reasons, the benefit of it, might be beneficial.

Generally, statistical linguistics use frequency rank rather than just frequency because it allows for easy comparison between different datasets. The frequency of the word "and" will be quite different in a 1 trillion word dataset vs. a 10 million word one. Of course, that doesn’t really matter here for Benford's Law analysis, but it does matter for cross-dataset comparison. We could do a Benford test on the frequency counts within each dataset (which, incidentally, we did in early stages of this work and everything looks pretty Benford-y), but we wouldn't be comparing apples-to-apples across datasets then. Using the frequency rank lets us say for a language overall, here's the rank of each word, and we can check every dataset with that same rank. 

This is an important point and we have added a more formal discussion of this point to the Methods and Data section.

User-Level Analysis on Social Media

The methodology here must be clarified. Is the analysis based again on the “frequency rank”?

I do not get if the conclusion related to the fact that “bot” social network accounts are less Benford’s law compliant comes from a visual inspection or if it is possible to have some statistical test to validate that point.

We have re-written this section to clarify the methodology. 

For the Parler accounts, we made the conclusion about bot status based on a human analysis of their posts. We have re-written this paragraph to make that clearer.

Conclusions:

If the author(s) decides to keep any of the comments on board, please revise the conclusion accordingly. Furthermore, I suggest that, given the quality of the results in the field, more academically informed conclusive remarks can be given on the areas covered: the linguistic aspect and the social network component.

We have updated the conclusion to include additional discussion along these lines.

General comments:

When one removes non-standard text, I refer, e.g. to “Other non-standard words are unlikely to appear in the Google corpra and will be ignored” there might be a “manipulation” of the data that somewhat connect to what is done in the case of accountancy when some figures are systematically changed. Therefore, extra care should be used. Some of those words might be neologisms, words that are outside current dictionaries but in use in “hurban” conversation; this is especially relevant for the component of the analysis related to social networks. My question would be, does consider words in official dictionaries or including those not-in-dictionaries would change the results drastically? If it is not possible to re-do the analysis, I understand. I am not necessarily asking for that, but a few comments on the impact on Benford’s law validity would help. This is partially there already when the %s are reported, but a clear statement where one says that these numbers are not enough for breaking Benford’s law may reassure a reader. For example in “Ausloos, M., Ficcadenti, V., Dhesi, G., & Shakeel, M. (2021). Benford’s laws tests on S&P500 daily closing values and the corresponding daily log-returns both point to huge non-conformity. Physica A: Statistical Mechanics and its Applications, 574, 125969” a point regarding the inclusion or exclusion of the “zeros” from the returns analysis is made when evaluating the second digit law.

The challenge with these words is that, since they aren't in the Google dataset for each language, they do not have a frequency rank. These do not make a difference to the analysis – we point out in the Additional Corpora section that they are <1% of the data. We agree that additional description of their impact on the validity is important and we have added that text along with a reference to the paper mentioned.

Check the usage of the words “corpora” and “corpus” I feel sometimes the usage of the words is interchanged, but the meaning is different.

Thanks for this – we had a consistent typo with "corpra" instead of "corpora" on top of all the other confusion. We were trying to use "corpora" simply as the plural of corpus. We hope this has been clarified in this version, but if we are unintentionally mis-using this still, we'd appreciate the feedback!

Congratulations on using genuine and non-genuine accounts for the analysis related to the bots! I really like it!

Thank you! We are so excited about these results!

---

## [Decision Letter · Decision Letter 1]

29 Aug 2023

Benford's Law Applies to Word Frequency Rank in English, German, French, Spanish, and Italian

PONE-D-23-08610R1

Dear Dr. Golbeck,

We’re pleased to inform you that your manuscript has been judged scientifically suitable for publication and will be formally accepted for publication once it meets all outstanding technical requirements. Please, a careful final reading is recommended to spot remaining typos.

Kind regards,

Luca De Benedictis, PhD

Academic Editor

PLOS ONE

**Comments to the Author**

Reviewer #1: All comments have been addressed

2. Is the manuscript technically sound, and do the data support the conclusions?

Reviewer #1: Yes

3. Has the statistical analysis been performed appropriately and rigorously? 

Reviewer #1: Yes

4. Have the authors made all data underlying the findings in their manuscript fully available?

Reviewer #1: Yes

5. Is the manuscript presented in an intelligible fashion and written in standard English?

Reviewer #1: Yes

6. Review Comments to the Author

Reviewer #1: Very good work, extensive and properly executed. A careful final reading is recommended to spot remaining typos.

7. PLOS authors have the option to publish the peer review history of their article (what does this mean?). If published, this will include your full peer review and any attached files.

Reviewer #1: No

---

## [Editor Report · Acceptance letter]

4 Sep 2023

PONE-D-23-08610R1 

Benford's Law Applies to Word Frequency Rank in English, German, French, Spanish, and Italian 

Dear Dr. Golbeck:

I'm pleased to inform you that your manuscript has been deemed suitable for publication in PLOS ONE. Congratulations! Your manuscript is now with our production department. 

Kind regards, 

on behalf of

Dr. Luca De Benedictis 

Academic Editor

PLOS ONE